**COMMUNICATIONS**

# Enantioselective radical conjugate additions driven by a photoactive intramolecular iminium-ion-based EDA complex

Zhong-Yan Cao[1], Tamal Ghosh[1] & Paolo Melchiorre [1,2]

The photochemical activity of electron donor–acceptor (EDA) complexes provides a way to generate radicals under mild conditions. This strategy has found application in chemical synthesis and recently in enantioselective catalysis. Reported methods classically relied on the formation of intermolecular EDA complexes, generated upon aggregation of two suitable reagents. Herein, we further expand the synthetic utility of this strategy demonstrating that an intramolecular EDA complex can trigger a photochemical catalytic enantioselective radical process. This approach enables radical conjugate additions to β-substituted cyclic enones to form quaternary carbon stereocenters with high stereocontrol using visible light irradiation. Crucial for success is the use of an amine catalyst, adorned with a carbazole moiety, which generates, upon condensation with enones, chiral iminium ions that show a broad absorption band in the visible region. This optical property originates from an intramolecular charge transfer π–π interaction between the electron-rich carbazole nucleus and the electron-deficient iminium double bond.

[1] ICIQ, Institute of Chemical Research of Catalonia - the Barcelona Institute of Science and Technology, Avinguda Països Catalans 16, 43007 Tarragona, Spain. [2] ICREA, Catalan Institution for Research and Advanced Studies, Passeig Lluís Companys 23, 08010 Barcelona, Spain. Correspondence and requests for materials should be addressed to P.M. (email: pmelchiorre@iciq.es)

The charge-transfer theory was formulated in 1952 by Robert Mulliken to rationalize the appearance of strong color on bringing together two colorless organic compounds[1]. This theory explains how the association of an electron-rich substrate (a donor **D**) and an electron-accepting molecule (an acceptor **A**) can induce the formation of a molecular aggregation in the ground state. This new intermediate, termed electron donor–acceptor (EDA) complex (Fig. 1a)[2], has physical properties that differ from those of the separated substrates. For example, its formation is generally accompanied by the appearance of a new absorption band, the charge-transfer band ($h\nu_{CT}$), which is associated with a transfer of a single electron (SET) from the donor to the acceptor. Often, the energy of this transition lies in the visible frequency range.

The photo-physics of EDA complexes have been extensively studied since the 1950s[3–5], while their use in chemical synthesis was exploited in a more intensive way only 20 years later[6–9]. Recently, the resurgence of visible light-driven processes has motivated chemists to reinvestigate the potential of EDA complex activation for promoting photochemical processes[10–24]. For example, our laboratories applied this strategy for enantioselective catalysis[25–28]. Specifically, we exploited the ability of a chiral catalyst to interact with a weakly polarized substrate (e.g. aldehydes and β-ketoesters) and transform it into an electron-rich chiral organocatalytic intermediate (such as enamines[25–27] or enolates[28]). This enabled the aggregation with an electron-poor reagent to form a colored EDA complex, whose photoactivity provided access to reactive open-shell intermediates (Fig. 1b). At the same time, the chiral organocatalytic intermediate ensured effective stereochemical control over the ensuing radical bond-forming process. In general, all the synthetic methods reported so far[6–28] relied on the excitation of intermolecular EDA complexes formed upon aggregation of two substrates/intermediates.

Here, we further expand the synthetic potential of this activation strategy demonstrating the possibility to form an *intramolecular* EDA complex and using its photochemical activity to drive a stereoselective radical reaction (Fig. 1c). Specifically, we used a chiral catalyst to generate an electron-poor intermediate upon condensation with a weakly polarized substrate. To induce the formation of an intramolecular EDA aggregation[29–32], the catalyst was adorned with an electron-rich moiety. The resulting intramolecular charge transfer π–π interaction elicited a broad absorption band in the visible region. Irradiation with visible light initiated a stereoselective radical process. To our knowledge, this study offers the first demonstration of the potential of photon-absorbing intramolecular EDA complexes in synthetic applications.

## Results

**Background**. The present study was motivated by our interest in developing enantioselective radical conjugate additions[33–36] via iminium ion activation[37]. This organocatalytic activation mechanism has found many applications for facilitating conjugate additions in ionic domains. In contrast, it was difficult to develop a stereoselective iminium ion trap of open-shell species. Recently, our laboratories reported a strategy, based on the combined action of a chiral organic catalyst **3**, bearing a redox-active carbazole moiety, and a photoredox[38] catalyst (**PC**), that enabled enantioselective radical conjugate additions to β-substituted cyclic enones **1** (Fig. 2a)[39,40]. During these studies, we isolated stable tetrafluoroborate salts of the chiral iminium ion **A-1**, generated upon condensation of a catalyst of type **3** (bearing a 3,6-di-*tert*-butyl-carbazole moiety) and an aliphatic enone (R′ = Me in **1**, Fig. 2b), which were characterized by X-ray single-crystal analysis. Surprisingly, these crystals showed an intense bright-yellow coloration (see UV–Vis spectrum in Fig. 2c). Typically, aliphatic iminium ions of type **A** can only absorb in the UV region[41,42] (below 350 nm, see for example, the absorption spectrum of iminium ion **A-2** in Fig. 2c).

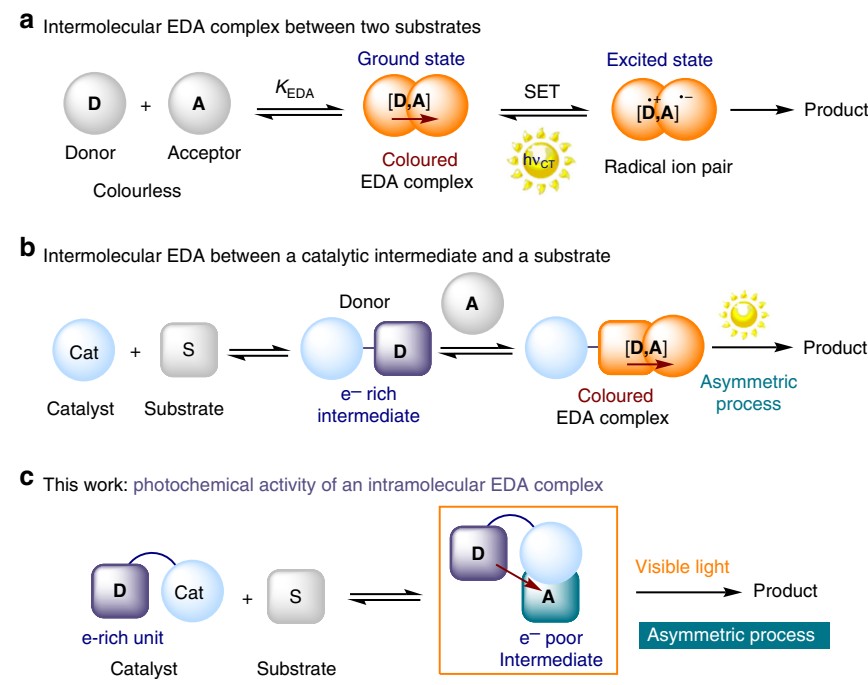

**Fig. 1** The EDA complex activation strategy for photochemical synthetic applications. **a** The principles of EDA complex activation and synthetic use enabled by light. **b** In situ generated catalytic intermediates can form an intermolecular EDA complex with a substrate. **c** A catalyst, adorned with an electron-rich unit, reacts with a substrate to generate an electron-poor intermediate prone to intramolecular EDA complex formation. $K_{EDA}$ association constant of EDA complex formation, **D** electron-rich molecule, **A** electron-poor molecule, SET single-electron transfer

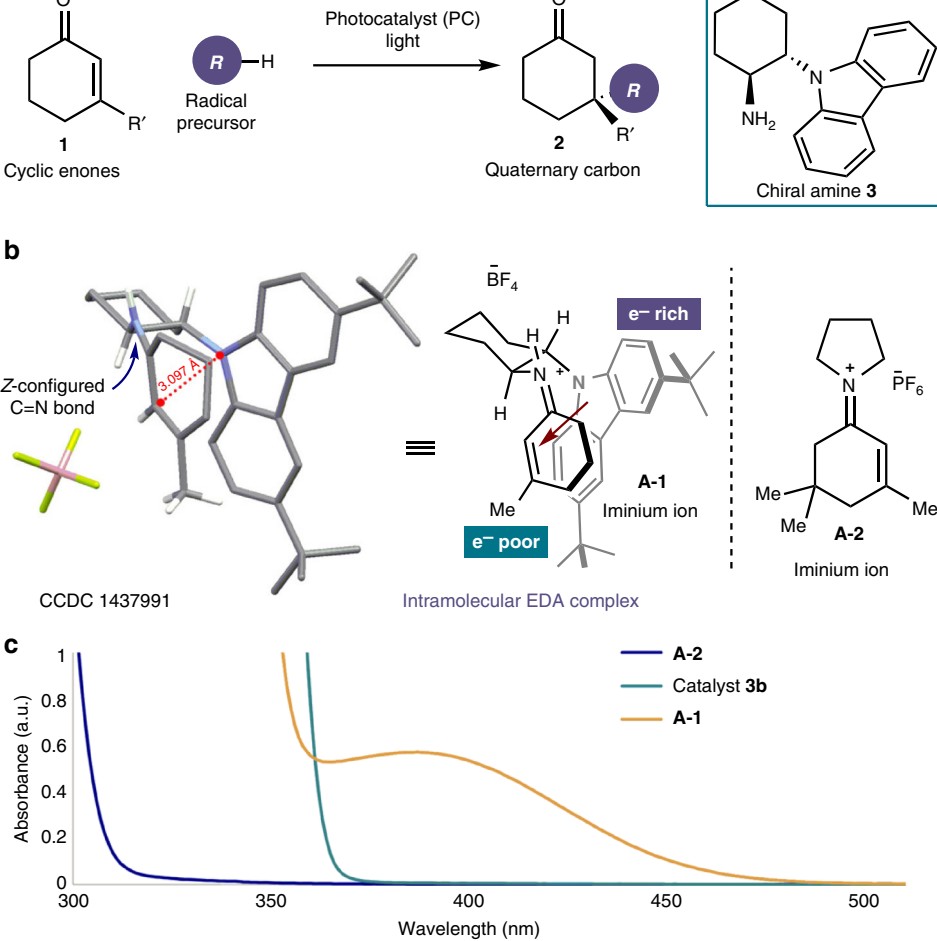

**Fig. 2** Background and initial insights. **a** Original strategy for the enantioselective iminium ion trapping of radicals based on the combination of a carbazole amine catalyst **3** and an exogenous photoredox catalyst (**PC**). **b** X-ray crystal structure of the carbazole-based iminium ion **A-1**. **c** UV–Vis spectra of the carbazole catalyst **3b** (structure in Table 1) and iminium ions **A-1** and **A-2** [1.0 mM] in CH₃CN

The X-ray structure of **A-1** provided a rationalization for its optical properties. The broad absorption band in the visible region is induced by a stabilizing *intramolecular* charge-transfer π–π interaction between the electron-deficient iminium ion double bond and the electron-rich carbazole nucleus. This type of intramolecular EDA[29–32] association generally results in a considerable redshift of the absorption bands. The intramolecular EDA complex also accounted for the interatomic separation between the carbazole nitrogen and the $sp^2$ α-carbon of the iminium ion (3.10 Å), which was significantly shorter than the van der Waals distance.

Given our interest in the photochemical activity and the synthetic utility of intermolecular EDA complexes generated from chiral organocatalytic intermediates[25–28,43], we wondered if the visible-light-absorbing properties of the iminium ion-based EDA complex **A** could be harnessed to design an enantioselective light-triggered process[44] *without the need for an external photoredox catalyst.*

**Design plan**. Figure 3 details our strategy for exploiting the photoactivity of the intramolecular iminium-ion-based EDA complex **A**, which is transiently generated in the ground state upon condensation of the carbazole aminocatalyst **3** with cyclic enone **1**.

Visible-light excitation of **A** would trigger an intracomplex SET from the electron-rich carbazole (the donor) to the iminium double bond (the acceptor), furnishing the chiral radical intermediate **B**. Using the photoexcitation of EDA complexes in chemical synthesis is not simple, mainly because of the unproductive, fast back-electron transfer (BET) which restores the ground-state EDA complex, making further reactivity difficult. Therefore, a central element of our approach was to identify a suitable mechanism to interrupt the BET process[45] between **B** and **A**. We envisioned that the long-lived carbazole radical cation in **B**, which is known to be a persistent species[46], could act as an effective oxidant. Specifically, we hoped that an SET would occur from an easily oxidizable electron-rich alkyl trimethylsilane **4** to **B**, furnishing the neutral 5π-electron β-enaminyl radical intermediate **C** and thus precluding the BET. This SET manifold would also generate the silyl radical cation **D**. The latter intermediate can readily undergo rapid desilylation in the presence of weak nucleophiles, including water[47,48], to form a carbon radical, which can then enter a radical conjugated addition manifold[39]. Importantly, after the stereoselective iminium ion radical trap, the electron-rich carbazole moiety would be positioned at a strategic position to promote a rapid intramolecular SET reduction of the unstable α-iminyl radical cation **E**[49]. This regenerates the carbazole radical cation within **F**, which would be able to oxidize a new molecule of the organic

**Fig. 3** Mechanistic proposal. Central to this study is the ability of the carbazole radical cation within **B**, generated upon visible-light excitation of the intramolecular EDA complex **A**, to drive the generation of radicals acting as an SET oxidant. SET single-electron transfer, BET back-electron transfer, Solv solvent, TMS trimethylsilyl

silane **4**. By regenerating the radical, this SET event would be the propagation step of a radical chain pathway. Hydrolysis of the neutral imine intermediate **G** would turn over catalyst **3** while providing product **2** bearing a quaternary carbon stereocenter.

**Optimization of the model reaction**. To validate our photochemical plan, we selected β-methyl cyclohexenone **1a** as the model substrate, while using the carbazole catalysts **3** to promote the formation of the chiral iminium ion **A** (Table 1). We performed the experiments in $CH_3CN$ at 35 °C under irradiation by a single high-power (HP) LED ($\lambda_{max} = 420$ nm) and using an irradiance of 15 mW/cm$^2$, which was controlled by an external power supply (full details of the illumination set-up are reported in Supplementary Figure 1). Water (2 equiv) was added to facilitate the generation of radicals upon desilylation of intermediate of type **D**. As radical precursors, we selected a pool of organic silanes **4a–c** with different oxidation potentials ($E_{ox}$ (**4**$^{·+}$/ **4**), as measured by cyclic voltammetry versus $Ag/Ag^+$ in $CH_3CN$). In consonance with the mechanistic requirement that radicals originate upon SET oxidation from the carbazole radical cation in **B**, only substrates **4** with comparable or lower oxidation potential than the catalyst carbazole unit should be suitable for reaction. This was the exact reactivity trend observed in the photochemical radical conjugate addition promoted by catalyst

**3a** (entries 1–3). Given the oxidizing power of the carbazole radical cation in **3a** ($E_{ox}$ (**3a**$^{·+}$/**3a**) = + 1.15 V), it is no surprise that only the trimethylsilyl carbazole **4a** ($E_{ox}$ (**4a**$^{·+}$/**4a**) = + 0.95 V) provided the desired conjugate addition product **2** bearing a quaternary stereocenter (entry 1). Substrates **4b** and **4c** remained completely unreacted. Substrate **4d**, having a dimethyl(phenyl) silyl group, provided better results than the trimethylsilyl (TMS) analog (speculatively because of the enhanced β-silyl stabilization of the intermediate radical cation of type **D**[50,51], compare entries 1 and 4), and was selected for further studies.

We then evaluated a family of chiral catalysts **3b–e** bearing different substitution patterns, which provided the redox-active carbazole unit with a range of electronic properties. As previously reported by our group[39,40], both enantiomers of catalysts **3** are accessible in good yields from readily available carbazoles through a five-step sequence (see Supplementary Note 1 in the Supporting Information for details on the synthesis of catalyst **3e**). In congruence with the mechanistic proposal, we found a correlation between the catalytic activity and the oxidation ability of the carbazole unit. The reaction was effectively promoted by catalysts bearing more electron-withdrawing groups (**3d** and **3e**), which magnified the oxidizing power of the carbazole radical cation (entries 7 and 8). In contrast, the presence of an electron-donating group in **3c** completely inhibited the process (entry 6). The best results in terms of reactivity and stereocontrol were

**Table 1 Optimization studies**

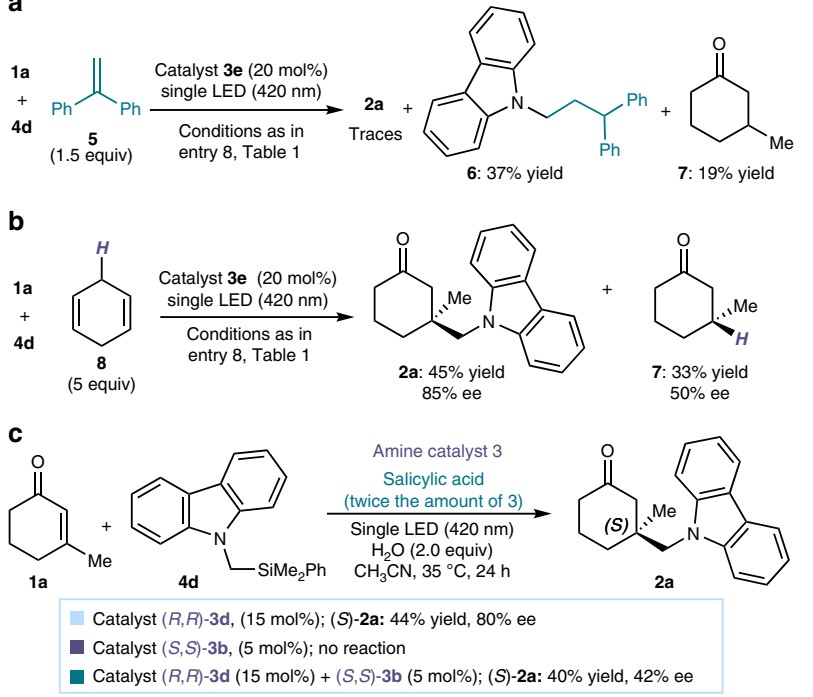

| Entry | Catalyst 3 | $E_{ox}$ (3$^{·+}$/3) (V)[a] | 4 | [%] Yield of 2[b] | [%] ee of 2 |
|---|---|---|---|---|---|
| 1 | 3a | +1.15 | 4a | 58 | 79 |
| 2 | 3a | +1.15 | 4b | 0 | – |
| 3 | 3a | +1.15 | 4c | 0 | – |
| 4 | 3a | +1.15 | 4d | 82 | 80 |
| 5 | 3b | +1.05 | 4d | 16 | 86 |
| 6 | 3c | +0.96 | 4d | 0 | – |
| 7 | 3d | +1.16 | 4d | 79 | 81 |
| 8 | 3e | +1.09 | 4d | 74 | 88 |

Reactions performed at 35 °C on a 0.1 mmol scale using 1.5 equiv. of **4** under illumination by a single high-power (HP) LED ($\lambda_{max}$ = 420 nm)
[a]$E_{ox}$ of the redox-active carbazole unit within catalysts **3a–e**, as measured by cyclic voltammetry vs. Ag/Ag$^+$ in CH$_3$CN
[b]Yield of isolated **2**

**Fig. 4** Mechanistic investigations. **a** Trapping the radical intermediate generated from **4d**. **b** The presence of an hydrogen donor **8** favors the formation of 3-methyl cyclohexanone **7**. **c** Entrainment experiment supporting a radical chain propagation mechanism

**Table 2 Substrate scope for the enantioselective trap of carbazole-derived radicals**

| Entry | 1 | | 4 | | | 2 | [%] Yield[a] | [%] ee |
|---|---|---|---|---|---|---|---|---|
| | **n** | **R¹** | **R²** | **R³** | **R⁴** | | | |
| 1 | 1 | Me(**1a**) | H | H | H | **2a** | 74 | 88 |
| 2 | | **1a** | Me | H | H | **2b** | 82 | 84 |
| 3 | | **1a** | OMe | H | H | **2c** | 82 | 87 |
| 4 | | **1a** | Ph | H | H | **2d** | 75 | 85 |
| 5[b] | | **1a** | *p*Me-Ph | H | H | **2e** | 95 | 87 |
| 6[b] | | **1a** | *p*Cl-Ph | H | H | **2f** | 87 | 87 |
| 7[b] | | **1a** | *2*-furyl | H | H | **2g** | 44 | 90 |
| 8[b] | | **1a** | *2*-thienyl | H | H | **2h** | 75 | 90 |
| 9 | | **1a** | H | Et | H | **2i** | 85 | 88 |
| 10 | | **1a** | H | F | H | **2j** | 78 | 86 |
| 11 | | **1a** | H | H | Me | **2k** | 82 | 88 |
| 12 | 1 | Et | H | H | H | **2l** | 76 | 67 |
| 13 | 1 | Bn | H | H | H | **2m** | 60 | 50 |
| 14 | 1 | ⊲ | H | H | H | **2n** | 78 | 56 |
| 15[c] | 0 | Me | H | H | H | **2o** | 45 | 74 |
| 16[c] | 2 | Me | H | H | H | **2p** | 74 | 95 |
| 17 | | **1q** | H | H | H | **2q** | 78 | 55 |
| 18 | | **1r** | H | H | H | **2r** | 68 | 0 |

Reactions performed over 48 h at 35 °C on a 0.1 mmol scale using 1.5 equiv. of **4** under illumination by a single high-power (HP) LED ($\lambda_{max}$ = 420 nm)
[a]Yield and ee measured on the isolated **2** (average of two runs per substrate)
[b]Reaction time: 72 h
[c]Benzoic acid (40 mol%) was used

**Table 3 Substrate scope for the enantioselective trap of α-silyl amine-derived radicals**

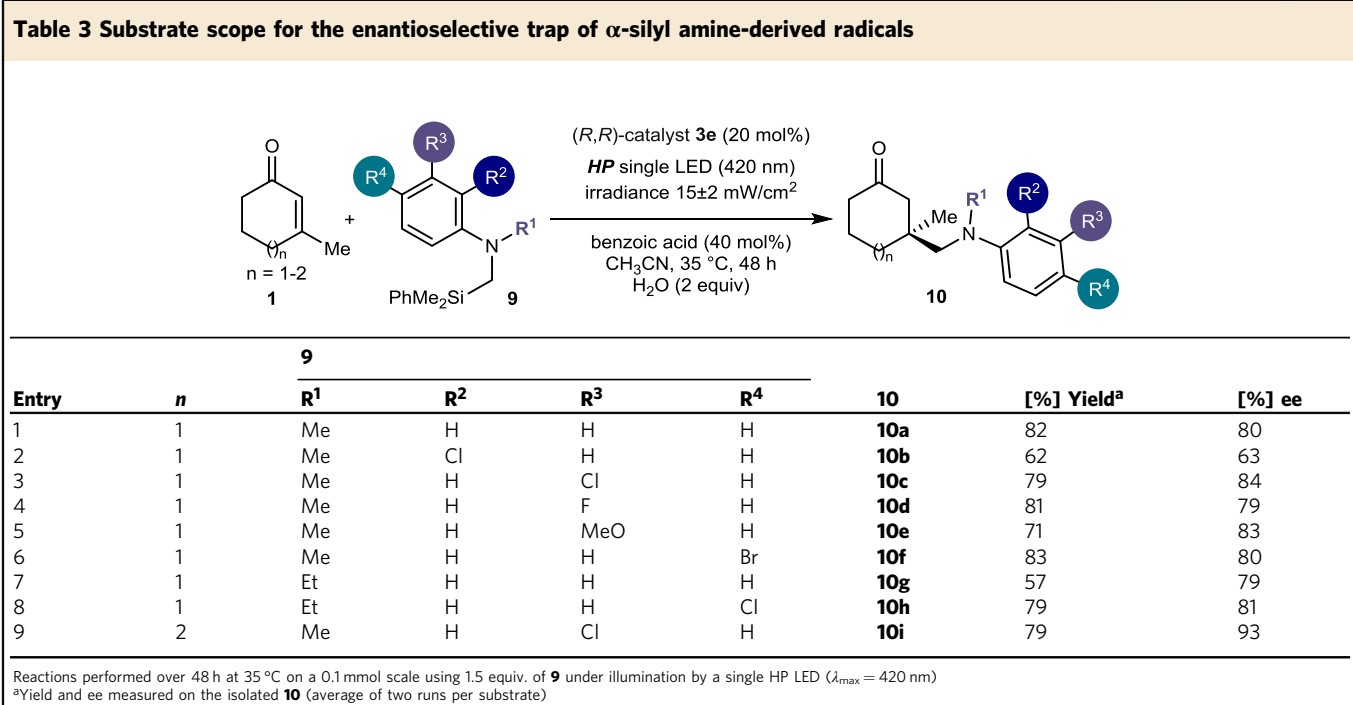

| Entry | n | 9 R¹ | R² | R³ | R⁴ | 10 | [%] Yield[a] | [%] ee |
|---|---|---|---|---|---|---|---|---|
| 1 | 1 | Me | H | H | H | **10a** | 82 | 80 |
| 2 | 1 | Me | Cl | H | H | **10b** | 62 | 63 |
| 3 | 1 | Me | H | Cl | H | **10c** | 79 | 84 |
| 4 | 1 | Me | H | F | H | **10d** | 81 | 79 |
| 5 | 1 | Me | H | MeO | H | **10e** | 71 | 83 |
| 6 | 1 | Me | H | H | Br | **10f** | 83 | 80 |
| 7 | 1 | Et | H | H | H | **10g** | 57 | 79 |
| 8 | 1 | Et | H | H | Cl | **10h** | 79 | 81 |
| 9 | 2 | Me | H | Cl | H | **10i** | 79 | 93 |

Reactions performed over 48 h at 35 °C on a 0.1 mmol scale using 1.5 equiv. of **9** under illumination by a single HP LED ($\lambda_{max}$ = 420 nm)
[a]Yield and ee measured on the isolated **10** (average of two runs per substrate)

provided by catalyst **3e**, which was adorned with substituents that can enhance both its steric shielding and oxidizing ability (entry 8).

**Mechanistic investigation**. Catalyst **3e** was selected for further investigations to better elucidate the mechanism. No product formation was detected in the absence of light or when replacing catalyst **3e** with cyclohexylamine, which lacked the redox-active carbazole core. Optical absorption studies (detailed in Supplementary Figure 4) indicated that only the in situ generated iminium ion **A** could absorb at 420 nm, the operative λ of the system. When performing the model reaction in the presence of ethene-1,1-diyldibenzene **5**, we observed the formation of adduct **6**, arising from the trap of the radical photogenerated from substrate **4d** (Fig. 4a). We also detected 3-methyl cyclohexanone **7**, a compound which likely arose from the β-enaminyl radical intermediate **C** (Fig. 3) upon hydrogen abstraction. The last observation suggests a possible pathway to turn over the catalyst **3** involved in the light-triggered radical generation manifold.

We then performed an additional experiment, where the model reaction was carried out in the presence of an excess of 1,4-cyclohexdiene **8**, a hydrogen donor that afforded **7** in a better yield and with moderate enantioselectivity (Fig. 4b). Control experiments established that catalyst **3e**, substrate **4d**, and light were all necessary for the formation of product **7**. Overall, these observations are congruent with the proposed mechanism for radical formation, where the photogenerated intermediate **B** oxidizes substrate **4** to furnish the radical along with intermediate **C**.

Finally, we conducted an experiment using two carbazole catalysts with an opposite absolute configuration (Fig. 4c). We mixed 15 mol% of (R,R)-**3d**, a catalyst that can efficiently promote the model reaction to give product (S)-**2a** in 80% ee, with 5 mol% of catalyst (S,S)-**3b**, which is completely inactive in these conditions. This catalyst mixture afforded the product (S)-**2a** with a greatly reduced enantioselectivity (42% ee) with respect to using **3d** alone. This result implies that, while only (R,R)-**3d** can be involved in the radical generation pattern, the subsequent iminium ion trap is governed by both catalysts, thus causing an

erosion of stereocontrol. This entrainment experiment (the non-active catalyst **3b** at initiation is instead apt at propagation) supports the mechanism proposed in Fig. 3, where the EDA-complex-based photochemical radical generation serves to initiate a radical conjugate addition sustained by a chain propagation mechanism. Our attempts to determine the quantum yield of the process, which would be useful to support the feasibility of a chain manifold, were frustrated by the impossibility of carefully establishing the photon flux of the high-power violet LED needed for the photochemical processes, a crucial requirement for reliably determining the quantum yield.

**Substrate scope and synthetic applications**. Adopting the optimized conditions described in Table 1, entry 8, we demonstrated the generality of the radical conjugate addition by evaluating a variety of cyclic enones **1** and α-carbazole silyl reagents **4** (Table 2). The resulting chiral products **2** contain a quaternary carbon center bearing a carbazole moiety, a structure found in optoelectronic materials, synthetic dyes, conducting polymers[52], and naturally occurring alkaloids[53].

A wide range of substituents at the carbazole core were well-tolerated, regardless of their electronic properties and position (products **2a–k**). Heterocycle rings, including furyl and thienyl moieties, could be included in the products (entries 7 and 8). As for the enones, a wide range of carbocycles and β-olefin substituents could be used. Ring systems that incorporate β-alkyl groups other than methyl reacted with a reduced enantioselectivity (entries 12–14), while the radical conjugate addition performed well for a diverse range of ring sizes, including cyclopentenyl and cycloheptenyl architecture (entries 15 and 16). Interestingly, linear enones reacted smoothly to provide the corresponding compounds **2q-r**, albeit with moderated or no stereoselectivity (entries 17–18). Crystals from compound **2a** were suitable for X-ray crystallographic analysis, which established the absolute configuration of the quaternary stereocenter.

The general applicability of a chemical strategy is crucial for evaluating its usefulness. We wondered if the photochemical

**Fig. 5** Expansion of the scope and synthetic utility. **a** Using different radical precursors than silanes. **b**, **c** Enantioselective synthesis of biologically relevant compounds

activity of the iminium-ion-based EDA complex could be expanded to successfully generate radicals from α-silyl amines 9[35,54–56], organic silanes with a low oxidation potential ($E_{ox} < 1.0$ V, see Supplementary Figure 12 for details). As detailed in Table 3, the resulting photochemical radical conjugate addition to cyclic enones efficiently afforded an array of adducts **10** with a quaternary stereocenter. In contrast, both α-indole and α-pyrrole silyl reagents remained unreacted under the optimal conditions (see Supplementary Figure 2).

Preliminary investigations indicated that our protocol is compatible with different radical precursors other than silanes. For example, N-phenylglycine **11**[57] ($E_{ox} = +0.42$ V vs. SCE in $CH_3CN$) and the (benzyloxy)methyl-substituted dihydropyridine **13** ($E_{ox} = +1.08$ V vs. $Ag/Ag^+$ in $CH_3CN$) provided the corresponding adducts **12** and **14**, respectively, with promising enantioselectivity (Fig. 5a). We also applied our method for the streamlined preparation of biologically relevant compounds. Figure 5b details a straightforward enantioselective two-step synthesis of compound **15**, which is potentially useful to prevent ophthalmic disease[58]. Last, we could stereoselectively prepare trans-alcohol **16**, the epimer of a compound useful for the treatment of influenza (Fig. 5c)[59].

In summary, we have demonstrated that an intramolecular EDA complex can be used to trigger a photochemical asymmetric radical process. Crucial for implementing the process was a chiral carbazole catalyst that do not contain any photosensitive unit, but rather it guides the photochemical formation of radicals by inducing the transient formation of visible-light-absorbing intramolecular iminium-ion-based EDA complexes. The method,

which avoids the need for external photoredox catalysts, offers the first example of synthetic applications triggered by the photochemical activity of an intramolecular EDA complex.

## Methods

**General**. For the exemplary reaction setup, see Supplementary Fig. 1. For more information on unreactive substrates, see Supplementary Fig. 2. For UV–Vis absorption studies, see Supplementary Figs. 3, 4. For cyclic voltammetry studies, see Supplementary Figs. 5–13. For the NMR spectra of compounds in this article, see Supplementary Figs. 14–47. For the product analysis with HPLC on chiral stationary phase, see Supplementary Figs. 48–80. For crystallographic information, see Supplementary Fig. 81 and the Supplementary Tables 1, 2, 3. For the synthesis of catalyst **3e** and substrates **2**, **4**, and **9**, see Supplementary Note 1. For the preparation of the iminium ion **A-2**, see Supplementary Note 2. For details of mechanistic studies, see Supplementary Note 3. For details of synthetic applications, see Supplementary Note 4. For general information, general experimental procedure, and analytic data of compounds synthesized, see Supplementary Methods. For additional references pertinent to the Supplementary Information, see Supplementary References.

**Catalytic enantioselective radical addition reaction to enones**. Exemplary, a 15 mL Schlenk tube was charged with the chiral carbazole-derived primary amine catalyst (R,R)-**3e** (0.04 mmol, 20 mol%), acid (0.08 mmol, 40 mol%, salicylic acid for substrate **4** and benzoic acid for substrates **9**), the organic silane **4** or **9** (0.15 mmol, 150 mol%), enone **1** (0.1 mmol, 100 mol%), $H_2O$ (0.2 mmol, 200 mol%), and 200 μL of $CH_3CN$. The mixture was placed under an atmosphere of argon, cooled with liquid nitrogen, and degassed via vacuum evacuation (5 min), backfilled with argon, and warmed to room temperature. The freeze-pump–thaw cycle was repeated three times, and then the Schlenk tube was placed into a support fitted with a 420 nm high-power single LED ($\lambda = 420$ nm). The irradiance was regulated at $15 \pm 2$ mW/cm², as controlled by an external power supply and measured using a photodiode light detector at the start and end of each reaction; the temperature was kept at 35 °C with a chiller connected to the irradiation plate (the setup is detailed in Supplementary Fig. 1). This setup secured a reliable irradiation and

temperature while keeping a distance of 1 cm between the reaction vessel and the light source. Stirring was maintained for the indicated time (generally 48 h), and then the irradiation was stopped. The reaction volatiles were removed in vacuum and the residue was purified by column chromatography to give the products **2** or **10** in the stated yield and enantiomeric purity. The reported yield and ee are average of two runs per substrate.

**Data availability**. The X-ray crystallographic coordinate for structures reported in this study have been deposited at the Cambridge Crystallographic Data Centre (CCDC) under deposition numbers CCDC-1819014 (**2a**). These data can be obtained free of charge from The Cambridge Crystallographic Data Centre via www.ccdc.cam.ac.uk/data_request/cif. The authors declare that all other data supporting the findings of this study are available within the article and Supplementary Information files, and also are available from the corresponding author upon reasonable request.

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

## Acknowledgements

Financial support was provided by the ICIQ, Agencia Estatal de Investigación (AEI, CTQ2016-75520-P and SEV-2013-0319), and the European Research Council (ERC 681840-CATA-LUX). Z.-Y. Cao thanks the EU for a Horizon 2020 Marie Skłodowska-Curie Fellowship (grant 702405).

## Author contributions

P.M. conceived the idea and supervised the whole project. Z.-Y. C. and T.G. designed and carried out the experiments. All authors discussed the results, contributed to writing the manuscript, commented on the manuscript, and approved the final version of the manuscript for submission.

## Additional information

**Competing interests:** The authors declare no competing interests.

