## [Peer Review File · Nature Communications]

Reviewers' Comments:

Reviewer #1:

Remarks to the Author:

In this study Melchiorre et al. demonstrate the photochemical addition of carbazole and alfa amino radicals to unsaturated cyclic ketones. The key finding is an intramolecular EDA complex generated from the condensation of an amine-based catalyst and an unsaturated cyclic ketone. The intramolecular EDA complex serves as a catalytic intermediate that facilitates the beta addition of a range of carbazol and aniline derivatives to form quaternary carbon stereocenters in high enantioselectivity. The mechanistic rational behind the intramolecular EDA complex is supported with a number of control reactions.

The finding is novel and can potentially be expanded to a range of other reactions and is of interest as it provides new tools for the synthesis of complex molecular architectures that can be used for the synthesis of APIs, etc. Several examples of this is also given within the body of the ms.

Furthermore, the mechanistic proposal is very exciting and in my opinion well investigated with several reactions that indicates the existence of several of the proposed catalytic intermediates.

The data analysis in the supporting information is performed in the in line with the requirements of the top journals in the field.

I have some minor things that could improve the state of the ms.

In figure S3 the color coding has disappeared please double check this image.

In connection to the amino-radical addition it would be appropriate to mention the work of Patrick Mariano who has performed much of the seminal work on alfa amino-radical additions to unsaturated aldehydes. The following reference should give some guidance to his work. Zhang, X. M.; Mariano, P. S. *J. Org. Chem.* 1991, 56, 1655–1660. On a side note, Marianos work indicates that there might be a considerable risk for a background reaction at irradiation at shorter wavelength that the authors circumvent by irradiating at 420 nm.

This is one of the best papers I have read this year and I would gladly recommend it for acceptance.

Reviewer #2:

Remarks to the Author:

The manuscript by Melchiorre and co-workers reports a new method for enantioselective conjugate radical additions relying on the formation of intramolecular EDA complexes to trigger SET upon visible-light excitation. The work represents the first example of intramolecular EDA complexes in synthetic applications. Given the topical interest in radical catalysis, and the high interest in the development of improved enantioselective radical processes using EDA complexes without the need for external photoredox catalysts, the work is certainly suitable for publication in *Nat. Commun.* Another interesting feature of this work involves preventing back electron-transfer, a process that hinders multiple approaches to radical chemistry, by using electron-rich alkyl-silanes. Although the enantioselectivity could be improved in several instances, it is expected that this work will open new prospects for enantioselective radical additions.

I only have a few minor comments:

1. The following paper illustrating the problem of back electron transfer in radical chemistry could

be cited: Chem. Rev. 2014, 114, 5959-6039.

2. A note on the availability of carbazole catalysts 3 should be given in the manuscript.
3. Please, comment on the origin of the inferior performance of 4a compared to 4d.
4. Please, comment if alpha-indole or alpha-pyrrole silyl reagents can be exploited in this chemistry.
5. Please, comment in the view of available oxidation potentials if simple alkyl silanes might become applicable in this approach.

Reviewer #3:

Remarks to the Author:

The article by Melchiorre deals with Enantioselective Radical Conjugate Additions Driven by the Photoactivity of an Intramolecular Iminium-ion-based EDA Complex. The authors use the traditional charge transfer chemistry to perform photochemical reaction. The article is just another incremental advancement from the same group who have shown that one can utilize this chemistry for radical reaction. The study did not add anything significant to the existing literature and does not pass the novelty test. In addition, in this reviewer's opinion this is an oversell of the chemistry presented which is well known to the synthetic community. For example in the introduction itself, the authors claim that "Over the last six decades, the photo-physics of EDA complexes have been extensively studied while their use in chemical synthesis has initially found limited application". This is absolutely a false claim as there are various reports in the last four decades on how to exploit charge transfer chemistry for chemical synthesis and even enantioselective synthesis (with light or radical initiation). Considering that this is an article, one of the aspects that needs to be included in the manuscript investigation is the spectroscopic investigations for such charge transfer process. This as well as prior publication from the same group does not have much detail on this aspect which is needed for such study. Over all this review is not in favor of publication. The study will be suitable for some specialized journal and not in an high impact journal like Nature or its sister journals.

Response to Reviewer 1

1. Reviewer 1 states: “The finding is novel and can potentially be expanded to a range of other reactions and is of interest as it provides new tools for the synthesis of complex molecular architectures that can be used for the synthesis of APIs, etc. [...] Furthermore, the mechanistic proposal is very exciting and in my opinion well investigated with several reactions that indicates the existence of several of the proposed catalytic intermediates. [...] This is one of the best papers I have read this year and I would gladly recommend it for acceptance.”

Our response: We thank the reviewer for the comments.

2. Reviewer 1 states: “I have some minor things that could improve the state of the ms. In figure S3 the color coding has disappeared please double check this image.”

Our response: The color coding in Figure S3, page S23 of the revised Supporting Information has been changed to facilitate visual understanding of the image.

3. Reviewer 1 states: “In connection to the amino-radical addition it would be appropriate to mention the work of Patrick Mariano who has performed much of the seminal work on alpha amino-radical additions to unsaturated aldehydes. The following reference should give some guidance to his work. Zhang, X. M.; Mariano, P. S. *J. Org. Chem.* 1991, 56, 1655–1660.”

Our response: we thank the reviewer for bringing this point to our attention. Patrick Mariano has certainly made great contributions in the area of SET-promoted photoadditions of amines to enones and we agree that his work has to be adequately acknowledged. Therefore, we have added three pertinent studies in the reference section: Reference 54 details a review article (*Acc. Chem. Res.* 1992, 25, 233-240) while References 55 and 56 include original pertinent studies (*JOC* 1991, 56, 1655 and *J. Am. Chem. Soc.* 1988, 110, 8099).

4. Reviewer 1 states: “On a side note, Marianos work indicates that there might be a considerable risk for a background reaction at irradiation at shorter wavelength than the authors circumvent by irradiating at 420 nm”.

Our response: the reviewer is right. As shown by Mariano (works cited in references 54-56, see point 3 above), the direct irradiation at shorter wavelength ($\lambda < 350$ nm) promotes a racemic α -amino radical addition reaction to cyclohexenones. Our strategy capitalizes upon the formation of the intramolecular iminium-ion-based EDA complex to disable this undesired background pathway. This is because the bathochromic shift induced by the EDA complex formation allows selective irradiation at 420 nm, where the direct excitation of the enone cannot take place (see Figure 2c in the main text for the UV spectra). The visible-light selective irradiation of the EDA complex is indeed crucial to achieve high enantioselectivity.

Response to Reviewer 2

5. Reviewer 2 states: “The manuscript by Melchiorre and co-workers reports a new method for enantioselective conjugate radical additions relying on the formation of intramolecular EDA complexes to trigger SET upon visible-light excitation..., the work is certainly suitable for publication in *Nat. Commun...*”

Our response: We thank the reviewer for the comments.

6. Reviewer 2 states: “The following paper illustrating the problem of back electron transfer in radical chemistry could be cited: *Chem. Rev.* 2014, 114, 5959-6039.”

Our response: The recommended reference has been added. Reference 45 now reads:

[45] Szostak, M., Fazakerley, N. J., Parmar, D. & Procter, D. J. Cross-coupling reactions using samarium(II) iodide. *Chem. Rev.* 114, 5959-6039 (2014).

7. Reviewer 2 states: “A note on the availability of carbazole catalysts **3** should be given in the manuscript.”

Our response: As per the reviewer’s request, the following note was added in page 3 of the revised manuscript, which reads:

“As previously reported by our group³⁹⁻⁴⁰, both enantiomers of catalysts **3** are accessible in good yields from readily available carbazoles through a five-step sequence (see section B in the Supporting Information for details on the synthesis of catalyst **3e**).”

8. Reviewer 2 states: “Please, comment on the origin of the inferior performance of **4a** compared to **4d**.”

Our response: We ascribe the higher reactivity of the dimethyl(phenyl) (Me_2Ph) silane substrate **4d** with respect to the TMS derivative **4a** to the enhanced β -silyl stabilization of the intermediate radical cation (JACS, **1990**, *112*, 1962). This effect leads to a more facile radical formation. A similar reactivity trend, where α -dimethylphenyl amines more efficiently produced the corresponding α -amino radicals than their TMS counterparts, was recently reported by Molander and coworkers (ACS Catal. **2017**, *7*, 6065).

We have added a sentence in page 3 to tentatively rationalize this reactivity trend, which reads: “Substrate **4d**, having a dimethyl(phenyl)silyl group, provided better results than the trimethylsilyl (TMS) analogue (speculatively because of the enhanced β -silyl stabilization of the intermediate radical cation of type **D**,^{50,51} compare entries 1 and 4), and was selected for further studies.” References 50 and 51 detail the studies mentioned above.

9. Reviewer 2 states: “Please, comment if alpha-indole or alpha-pyrrole silyl reagents can be exploited in this chemistry.”

Our response: We synthesised four different α -indole silyl reagents bearing either electron-donor or electron-poor substituents; unfortunately, all of them remained completely unreacted under the optimized photochemical conditions (see Figure below). As for the pyrrole, we tested the reactivity of 1-((dimethyl(phenyl)silyl)methyl)-2,5-dimethyl-1H-pyrrole, but no reaction took place. The absence of reactivity, in spite of the apparently suitable oxidation potentials of these substrates, remains unclear.

10. Reviewer 2 asks: “Please, comment in the view of available oxidation potentials if simple alkyl silanes might become applicable in this approach.”

Our response: Simple alkyl silanes possess oxidation potentials generally higher than +2.0 V (see for example Yoshida et al, JACS, **1990**, *112*, 1962, cited as Reference 50 in the revised manuscript). For example, the reported oxidation potential for $\text{C}_7\text{H}_{15}\text{CH}_2\text{SiMe}_2\text{Ph}$ is as high as +2.25 V versus Ag/Ag^+ in MeCN , thus making the SET oxidation of this substrate unfeasible under our reaction conditions (the carbazole radical cation has an $E_{\text{ox}}(\mathbf{3e}^+/\mathbf{3e}) = +1.09$ V). In line with this reasoning, the results in Table 1 show that even silanes **4b** and **4c**, which are relatively easy to oxidize ($E_{\text{ox}}(\mathbf{4}^+/\mathbf{4}) = +1.74$ V and +1.49 V, respectively) remained completely unreacted (entries 2 and 3).

Response to Reviewer 3

11. Reviewer 3 states: “The article by Melchiorre deals with Enantioselective Radical Conjugate Additions Driven by the Photoactivity of an Intramolecular Iminium-ion-based EDA Complex. The authors use the traditional charge transfer chemistry to perform photochemical reaction. The article is just another incremental advancement from the same group who have shown that one can utilize this chemistry for radical reaction. The study did not add anything significant to the existing literature and does not pass the novelty test.”

Our response: We thank the referee for the time dedicated to review the manuscript. Our view of the novelty introduced by this study is much more aligned with the opinions expressed by reviewers 1 and 2, who deemed this work as interesting, mainly for the conceptual implications. To our knowledge, this method offers the first example of synthetic applications triggered by the photochemical activity of an intramolecular EDA complex, a concept that can find other applications in synthetic chemistry.

12. Reviewer 3 states: “In addition, in this reviewer’s opinion this is an oversell of the chemistry presented which is well known to the synthetic community. For example in the introduction itself, the authors claim that “Over the last six decades, the photo-physics of EDA complexes have been extensively studied while their use in chemical synthesis has initially found limited application”. This is absolutely a false claim as there are various reports in the last four decades on how to exploit charge transfer chemistry for chemical synthesis and even enantioselective synthesis (with light or radical initiation).”

Our response: our intention was to explain that, while the photophysical properties of EDA complexes have attracted a great deal of attention, their synthetic potential was exploited to a less extent. We never claimed that EDA complex activation was not used in the last 40 years, and surely my intention was not to oversell the chemistry or to make “absolutely false claims”.

It seems that our initial explanation was not clear, therefore we have reworded part of the introduction, including an additional reference (Ref. 7) to better detail the early use of EDA complexes in synthetic photochemistry (besides the originally included examples). The new paragraph in the introductory section now reads: “The photo-physics of EDA complexes have been extensively studied since the 1950s³⁻⁵, while their use in chemical synthesis was exploited in a more intensive way only 20 years later⁶⁻⁹. Recently, the resurgence of visible light-driven processes has motivated chemists to reinvestigate the potential of EDA complex activation for promoting photochemical processes¹⁰⁻²⁴.”

13. Reviewer 3 states: “Considering that this is an article, one of the aspects that needs to be included in the manuscript investigation is the spectroscopic investigations for such charge transfer process. This as well as prior publication from the same group does not have much detail on this aspect which is needed for such study.”

Our response: I do not fully understand the concerns of the reviewer. Already in the original manuscript and in the Supporting Information, we reported spectroscopic investigations (Figure 2c) to support the formation of an EDA complex. We could not perform other classical spectroscopic studies (as we usually do, see Refs 13 and 27, for examples), including the Job’s method of continuous variations or the Benesi–Hildebrand and the Foster methods for the determination of the association constant (K_{EDA}), for the simple reason that, in this specific case, we dealt with an *intramolecular* EDA complex. This naturally complicates matters. However, we could provide direct and unambiguous evidence for EDA complex formation by means of X-ray crystallographic analysis, *which is unanimously considered as the method of excellence to provide compelling structural and mechanistic information*. Since the reviewer did not specify any spectroscopic studies that could be useful to characterize the EDA complex further, we are not able to further address this issue.

I hope these changes meet with your approval. Please do not hesitate to contact me if you have further comments or require further information.

I look forward to hearing from you.